# Soil Microbial Communities and Wine Terroir: Research Gaps and Data Needs

**DOI:** 10.3390/foods13162475

**Published:** 2024-08-06

**Authors:** Gabriela Crystal Franco, Jasmine Leiva, Sanjiev Nand, Danica Marvi Lee, Michael Hajkowski, Katherine Dick, Brennan Withers, LuzMaria Soto, Benjamin-Rafael Mingoa, Michael Acholonu, Amari Hutchins, Lucy Neely, Archana Anand

**Affiliations:** 1Department of Biology, College of Science and Engineering, San Francisco State University, 1600 Holloway Avenue, San Francisco, CA 94132, USA; gfranco3@mail.sfsu.edu (G.C.F.); jleiva2@sfsu.edu (J.L.); snand1@mail.sfsu.edu (S.N.); dlee41@sfsu.edu (D.M.L.); mhajkowski@mail.sfsu.edu (M.H.); kdick@mail.sfsu.edu (K.D.); bwithers@sfsu.edu (B.W.); lsoto@sfsu.edu (L.S.); bmingoa@sfsu.edu (B.-R.M.); macholonu@sfsu.edu (M.A.); 2Department of Biology, Howard University, 2400 6th St NW, Washington, DC 20059, USA; amari.hutchins@bison.howard.edu; 3Neely Winery, Spring Ridge Vineyard, 555 Portola Road, Portola Valley, CA 94028, USA; lucy@neelywine.com

**Keywords:** wine, vineyard, terroir, soil, microbial composition, function

## Abstract

Microbes found in soil can have a significant impact on the taste and quality of wine, also referred to as wine terroir. To date, wine terroir has been thought to be associated with the physical and chemical characteristics of the soil. However, there is a fragmented understanding of the contribution of vineyard soil microbes to wine terroir. Additionally, vineyards can play an important role in carbon sequestration since the promotion of healthy soil and microbial communities directly impacts greenhouse gas emissions in the atmosphere. We review 24 studies that explore the role of soil microbial communities in vineyards and their influence on grapevine health, grape composition, and wine quality. Studies spanning 2015 to 2018 laid a foundation by exploring soil microbial biogeography in vineyards, vineyard management effects, and the reservoir function of soil microbes for grape-associated microbiota. On the other hand, studies spanning 2019 to 2023 appear to have a more specific and targeted approach, delving into the relationships between soil microbes and grape metabolites, the microbial distribution at different soil depths, and microbial influences on wine flavor and composition. Next, we identify research gaps and make recommendations for future work. Specifically, most of the studies utilize targeted sequencing (16S, 26S, ITS), which only reveals community composition. Utilizing high-throughput omics approaches such as shotgun sequencing (to infer function) and transcriptomics (for actual function) is vital to determining the specific mechanisms by which soil microbes influence grape chemistry. Going forward, understanding the long-term effects of vineyard management practices and climate change on soil microbiology, grapevine trunk diseases, and the role of bacteriophages in vineyard soil and wine-making would be a fruitful investigation. Overall, the studies presented shed light on the importance of soil microbiomes and their interactions with grapevines in shaping wine production. However, there are still many aspects of this complex ecosystem that require further exploration and understanding to support sustainable viticulture and enhance wine quality.

## 1. Introduction

The concept of terroir in wine refers to the unique combination of environmental factors, including soil, climate, topography, and human practices, that influence the characteristics of grapes and, ultimately, the flavor and quality of wine [1,2]. While terroir is traditionally associated with macro-level factors such as climate, topography, physical and chemical soil characteristics, recent research has highlighted the role of microbiota, specifically grapevine-associated microbial communities, in potentially shaping the terroir effect [3].

Grapevines host a diverse array of microorganisms, including bacteria, yeasts, and fungi, on the surface of the grapes, within the grapevine itself, and the bulk soil [4,5]. These microbial communities can vary significantly between vineyards, regions, and even individual vines [6]. They play a crucial role in vineyard ecology, interacting with the plant and influencing its growth [7], health, and the development of grapes. For example, the grape-associated yeast community is a vital component of the vine–wine system contributing to terroir [8]. Additionally, the diversity and proportion of yeast species change with the grape’s maturation stage. As grapes begin to ripen, they are predominantly inhabited by basidiomycetous yeasts [9]. As maturation continues, these initial colonizers are replaced by ascomycetous species that exhibit oxidative or weak fermentative properties, including *Hanseniaspora*, *Metschnikowia*, *Pichia*, and *Candida* [9]. Notably, *Saccharomyces cerevisiae*, the primary yeast responsible for wine fermentation, is infrequently observed. In contrast, overmatured, damaged, or botrytized grapes favor the growth of yeasts with robust fermentative characteristics and others like *Pichia*, *Zygoascus hellenicus*, *Zygosaccharomyces*, and *Torulaspora* [9,10].

The microbiota associated with grapevines can influence terroir in several ways, based on which they can be classified into three categories:

Soil microbiota: Soil is an essential component of terroir, and the microbial communities within the soil can likely impact vine health and grape characteristics. Microbes in the soil interact with the vine’s root system, affecting nutrient availability, water uptake, and overall vine physiology. Links to the production of metabolites that influence grapevine metabolism and flavor compounds in the grapes are suggested in some literature but not conclusive [11,12,13]. Additionally, knowledge concerning variability within and between vineyards and regions and their contribution to wine terroir is still fragmented.

Epiphytic microbiota: Microorganisms present on the grape, leaf, and bark surfaces are the epiphytic microbiota. To date, the literature suggests that the composition of epiphytic microbiota can be influenced by vineyard management practices, such as the use of pesticides or fungicides [1,11,14]. However, the extent to which epiphytic microbiota can affect the fermentation process and contribute to the sensory attributes of the resulting wine is largely unknown. Recent research has begun to indicate that yeasts and bacteria on grape skins can influence the initiation and progression of fermentation, leading to different flavor profiles [15].

Endophytic microbiota: Endophytes are microorganisms that live within the tissues of grapevines. These microbes can have various effects on the vine, including enhancing nutrient uptake, modulating the plant’s immune system, and producing bioactive compounds. The presence and diversity of endophytic microbiota can vary between grape varieties and vineyard sites, contributing to the unique terroir expression as suggested by Compant et al. [16], Pacifico et al. [17], and Hamaoka et al. [18], to name a few.

Among the studies to date on the influence of microbial communities on wine terroir, the contribution of the soil microbiome (in comparison to epiphytic and endophytic microbiota) remains inconclusive and least scientifically explored. To address this gap, we review the current state of knowledge of soil microbiota contribution to terroir expression. We present research gaps and highlight future areas of research that warrant attention. For the purpose of this paper, we focus on literature that specifically looks at microbial communities in the soil.

Although understanding the influence of soil microbial communities on wine production is crucial, it is also important to explore how microbial biogeography and activity might respond to climate changes. By studying the complex interactions between soil microorganisms and the environment, we can gain valuable insights into the role of soil microbiota in shaping terroir. This knowledge allows us to comprehend how soil microbial communities contribute to the unique characteristics and flavors found in wines, ultimately helping us manage and manipulate these communities to enhance desired terroir traits or preserve the distinctiveness of specific terroirs. Therefore, in addition to reviewing the literature on soil microbiota in vineyards, we also discuss vine management practices, including disease and topics of interest such as bacteriophages and their likely relationship with soil microbiota and wine terroir.

## 2. Current State of Knowledge on the Microbiota Contribution to Terroir Expression

We conducted a PubMed literature search using the keywords “soil”, “microbial communities”, and “wine terroir”, resulting in 24 studies published between 2015 and 2023 (Table 1) and five review papers [3,19,20,21,22]. Studies focusing on grapevine microbiomes in plant parts, rather than bulk soil, were excluded from this study.

### 2.1. The “Core Microbiome” Concept

Early studies from 2015 to 2018 [13,23,24,25,26,27,28,29,30,31] laid the foundation by exploring soil microbial biogeography in vineyards, vineyard management effects, and the reservoir function of soil microbes for grape-associated microbiota. For instance, Zarronaindia et al. [13] mention that belowground bacterial communities differed significantly from those aboveground, and yet the communities associated with leaves, flowers, and grapes shared a greater proportion of taxa with soil communities than with each other, suggesting that soil may serve as a bacterial reservoir. Additionally, the authors define “core” microbiota found belowground that included those associated with *Bradyrhizobium*, *Steroidobacter*, and *Acidobacteria* species. Mezzasalma et al. [27] shared that the grape microbiome could be influenced by a “common microbiome” influenced by farming practices and climate conditions. This was deduced by observing microbes present at harvest and prior to fermentation that were commonly observed in soil bacteria *Pasteurellales* and *Bacteroidales* as well as *Rhodospirillales* and *Lactobacillales*. The studies point out the consistent presence of common soil bacteria as a source of microorganisms in grapefruit; however, we speculate that this could be driven by rain splash-, wind-, and insect-driven microbial migration [11].

Although there is limited information specifically about bulk soil microbes and their connection to wine terroir or grapefruit microbiomes, later studies (2019 onwards) appear to have a more specific and targeted approach, delving into the relationships between soil microbes and grape metabolites, the microbial distribution at different soil depths, and microbial influences on wine flavor and composition. Darriaut et al. [46] discussed microbes found in vineyard soils that could potentially influence terroir (*Actinobacteria* was prevalent in vineyard soils and linked to the production of secondary metabolites that may influence grape and wine characteristics; *Proteobacteria* was commonly found in vineyard soils and associated with nutrient cycling and plant growth promotion, potentially affecting grape quality and wine terroir. The same study mentioned that *Enterobacter* and *Paenibacillus* were observed to promote plant growth and cause metabolic changes when added to the soil. While not specifically linked to wine terroir, these changes could potentially influence grape characteristics. Beule et al. [47] discuss the abundance, diversity, and function of soil microorganisms in temperate agroforestry systems, which could be relevant to vineyard soil microbiology. Liu et al. [35] and Oyuela Aguilar et al. [36] examined both bacterial and fungal communities at different soil depths. However, specific findings from these studies in connection with wine terroir are not provided in the search results. Regecová et al. [43] observed a dominating presence of lactic acid bacteria during fermentation, specifically the *Lactiplantibacillus*, *Leuconostoc*, *Oenococcus*, and *Pediococcus* genera. These genera contain species associated with malolactic fermentation, which results in acidity reduction while enriching the wine with diacetyl and esters compounds. Regecová et al. [43] also noted a close relationship between population densities of lactic acid bacteria and non-*Saccharomyces* yeasts and the concentrations of metabolites such as biogenic amines, including histamine and tyramine. Notably, there were overlapping yeast species found in soil, leaf, and grape berries (*Metschnikowia pulcherrima*), also validating the “core microbiota” theory that came up in the literature from earlier years.

### 2.2. Dominant Groups in Vineyard Soil Microbial Communities

Concerning bacterial populations (Figure 1), twelve studies found *Proteobacteria* to be the most dominant phylum present [13,23,26,28,30,32,33,35,36,38,39,42], majority of which also found *Actinobacteria* to be present in high abundance. Ten studies noted *Acidobacteria* to be present in high abundance [13,23,26,28,32,33,35,36,38,42], while eight studies identified the presence of *Bacteroidetes* in high abundance [13,23,26,28,30,35,36,42]. *Gemmatimonadetes* was noted in six studies, though not as dominant as the aforementioned phyla [26,28,32,35,39,42], eight studies identified *Firmicutes* as one of the present phyla [13,26,28,30,33,35,38,42], and ten studies identified *Planctomycetes* in a mix of high, medium, and low abundance [13,23,26,28,30,32,36,38,39,42].

Commonly observed fungal phyla were *Ascomycota*, *Basidiomycota*, *Chytridiomycota*, *Mucoromycota*, and *Clomeromycota*. Ten studies found *Ascomycota* in high abundance and, in some cases, the most abundant [26,28,30,31,32,33,35,36,39,40]. *Basidiomycota* was observed in ten studies, although in high, medium, and low abundance [26,28,30,31,32,33,35,36,39,40]. *Chytridiomycota* and *Mucoromycota* were observed but not as dominant as the other phyla—Basidiomycota and Ascomycota [28,30,35,36,40]. *Glomeromycota* was observed but in low abundance [32,35,36].

### 2.3. Contributing Factors That Link Soil-Associated Microbiota to Wine Terroir

While there is growing evidence supporting the significance of soil microbial influence on grapevines and their associated microbiota, aspects of this interaction remain subject to debate and further investigation. For example, Zarraonaindia et al. [13] focused on the spatial and temporal dynamics of bacterial communities associated with grapevine organs (leaves, flowers, grapes, and roots) and soils. The study explored factors like vine cultivar, edaphic parameters, vine developmental stage, and vineyard that influence the microbial communities, but it did not directly address the influence of soil on grape microbiota. On the other hand, Chou et al. [29] investigated the impact of under-vine soil management practices (herbicide application, soil cultivation, and natural vegetation) on the microbiomes of soil and grapes in a Riesling vineyard. The study showed that soil management practices influenced the soil microbiome but did not have corresponding changes in the grape-associated microbiome, suggesting that other vineyard management practices or environmental factors may be more influential in shaping the grape microbiota.

The PubMed search also resulted in 5 reviews. We discuss them briefly in this section: Belda et al. [3] highlight the underestimated role of the soil microbiome in wine production. The study reveals that the soil-associated microbiota is likely to influence soil chemistry, grapevine health, and the final sensory properties of wines, calling for a deeper understanding of these crucial interactions for precision enology practices. Liu et al. [35] emphasize the role of microbial biogeography, shaped by geographical, climatic, and viticultural factors, as a new perspective to enhance regional characteristics and optimize wine production by managing the present microbes. Relevant to the current study, a review [20] highlights that the role of region-specific microbial communities (microbial terroir) in defining wine characteristics is still debated, requiring further research for a clearer understanding. Cobos et al. [21] discuss how the grapevine microbiome offers potential sources for new and promising biocontrol agents that could serve as effective tools in controlling grapevine trunk diseases. Lastly, Wei et al. [22] discuss the benefits of mimicking natural ecological cultivation to enhance microbial diversity and sustainability in large-scale natural wine practices. Together, these studies highlight the complex interactions between soil microbiomes, grapevines, and the production of high-quality wine. However, there are still many aspects of this complex ecosystem that require further exploration and understanding to support sustainable viticulture and enhance wine quality. Therefore, in the following sections, we discuss contributing factors that link soil-associated microbiota to wine terroirs, such as vineyard management, diseases, phages, and next-generation sequencing as topics that are important drivers of our knowledge of the changing landscape of soil microbes in vineyards.

#### 2.3.1. Climate Change Impacts

Investigating the effects of climate change on microbial community dynamics, their functional roles, and their implications for wine production can help prepare the wine industry for potential challenges [38]. Notably, one of the few papers that discusses soil diversity impacts through climatic condition changes was Rivas et al. [38]. The study indicates a consistent set of microorganisms in both soil and wine (*Proteobacteria*, *Actinobacteria*) that remain steady over multiple vintage years from Argentina. Consistently, these two groups were also documented (Table 1) as the most dominant phyla in the other short-term studies from 2015–2023 that were included in this perspective.

#### 2.3.2. Vineyard Management Practices

Vineyard management practices have profound impacts on both the immediate and long-term health of vineyards, influencing grapevine productivity and overall ecosystem sustainability, with soil health being a key concern [48]. The vineyard management techniques discussed in the literature (Table 1) are irrigation management [34,36], canopy management [34], pest and disease control [27,43], nutrient management [23,30], precision viticulture [37], and soil tilling [28]. Liu et al. [35] compares organic and conventional vineyard management practices and their effects on soil health and microbial diversity. Vineyard managers use techniques like cover cropping, composting, and organic fertilizers to maintain soil health, which not only ensures grapevines’ viability but also contributes to the sustainability of the vineyard ecosystem. Understanding these effects is crucial for developing environmentally friendly and economically viable viticultural methods. Although the studies focus on the effects of some vineyard management practices, the emphasis is not on providing in-depth descriptions of the practices and mechanisms of influence. To fully grasp the implications of various agricultural approaches on soil biodiversity and vineyard microbiomes, further research is warranted.

#### 2.3.3. Comparative Arguments on Soil Tilling

In addition to investigating vineyard management effects locally, comparative studies spanning different viticultural regions worldwide can offer valuable insights [49]. Specifically, soil tilling in viticulture is likely to have significant implications on nutrient and soil organic carbon content with correlations to the microbial community composition and associated function [50]. Yet, the impact of this practice on soil microbial abundance and richness, as well as its link to wine terroir, remains fragmented and contradictory. Andrade et al. [51], Buckley & Schmidt [52], and Jangid et al. [53] observed that no-till fields tended to have higher microbial diversity compared to tilled fields. However, the studies report highly variable results on the number and type of unique operational taxonomic units (OTUs) in tilled vs. no-till fields. *Pseudomonas*, *Bacillus*, *Streptomyces*, *Rhizobium*, and *Actinobacteria* in Andrade et al. [51], vs. *Acidobacteria*, *Proteobacteria*, *Firmicutes*, and *Bacteroidetes* in Buckley & Schmidt [52] and *Nitrosospira*, *Sphingomonas*, *Flavobacterium*, *Burkholderia*, and *Mycobacterium* in Jangid et al. [53]. Such cross-regional analyses have the potential to reveal both common patterns and unique characteristics associated with specific wine-producing areas.

#### 2.3.4. Precision Viticulture

Advancements in precision viticulture offer significant potential for targeted microbial management in vineyards [54]. However, among the 24 studies listed, only Teixeira et al. [37] directly discuss precision viticulture by employing advanced molecular techniques to enhance vineyard management. Their study employed DNA-based assays to detect single nucleotide polymorphisms (SNPs) in three genes of the anthocyanin pathway (UFGT, F3H, and LDOX), demonstrating how advanced molecular techniques can be integrated into vineyard management strategies. It is worth pointing out that the study appears to be more focused on the genetics of the grapevine itself rather than soil microbes. Other studies, while not explicitly labeled as precision viticulture, may involve practices and methodologies that align with the principles of precision viticulture, such as detailed monitoring and management of soil microbial communities in Hendgen et al. [28], Oyuela Aguilar et al. [36], and Gobbi et al. [42].

### 2.4. Future Directions for Soil Microbial Communities and Wine Terroir Research

Despite the growing body of research implying connections between soil microbes and wine terroir, the precise mechanisms underlying these interactions remain elusive. Next-generation sequencing (NGS) has emerged as a promising tool to unravel these complexities, offering detailed insights into microbial diversity and function. However, additional approaches such as precision viticulture, bacteriophages, and grapevine trunk disease management also hold the potential to enhance our understanding of these interactions. This section explores these future research areas that could provide a deeper understanding of soil microbial communities and their impact on wine terroir.

#### 2.4.1. Next-Generation Sequencing

To identify and quantify the microorganisms present in vineyard soil (Table 1), several studies have opted to use next-generation sequencing. Among the studies we reviewed, 16 focused on amplicon sequencing of the 16S rRNA gene, which is widely used in microbiology for the identification and classification of microorganisms, particularly bacteria and archaea. Twelve papers focused on sequencing the ITS regions and their subregions, and two papers used 26S rDNA. ITS and 26S rDNA are used widely to characterize eukaryotic organisms. Relevant to wine making, yeast communities were of interest to these papers. Only two studies utilized shotgun sequencing that analyzed entire genomes and complex microbial communities without the need for prior knowledge of specific DNA regions.

Although targeted sequencing (16S rRNA, ITS, and 26S rDNA) can reveal insights into the microbial community composition, shotgun sequencing offers a broader picture of the entire genome, making it suitable for functional information as well. However, the information obtained from shotgun sequencing (also sometimes referred to as whole genome sequencing or metagenomics, derived from DNA) can only lead to inferences about function. On the other hand, transcriptomics (derived from RNA rather than DNA) provides insights into the active metabolic pathways and biological processes occurring in the soil, giving a more dynamic view of microbial activity than just analyzing the microbial composition. To our knowledge, no studies have utilized transcriptomics to investigate the actual function of soil microbial communities in vineyard settings.

Metagenomics and transcriptomics of soil microbial communities in vineyards offer a powerful tool for gaining functional insights into soil microbiomes, supporting sustainable vineyard management, and contributing to the production of high-quality wines with a distinct terroir. Firstly, it helps predict microorganisms’ roles in nutrient cycling, organic matter decomposition, and other essential processes for vineyard health [55]. Secondly, transcriptomics helps in monitoring how soil microbial communities respond to changes in environmental factors, such as climate, soil management practices, and agricultural inputs [56]. Thirdly, this type of analysis can help identify specific microbial species or groups that play essential roles in promoting soil health, enhancing nutrient availability, and protecting grapevines from diseases. Such beneficial microbes can be targeted for potential use as biofertilizers or biocontrol agents. Lastly, such methods help in understanding how soil microbes contribute to the regional identity of wines (terroir), which is essential for promoting authenticity and quality [57].

#### 2.4.2. Expanding Precision Viticulture

The integration of cutting-edge technologies into viticulture can optimize soil microbial communities, bolster grapevine health, and elevate overall wine quality. Delving into the potential of precision viticulture to influence the dynamics of vineyard microbiomes can lead to innovative practices that maximize wine production efficiency while maintaining environmental sustainability. For example, integrating multispectral, hyperspectral, and thermal sensing (which are among the most widely used sensors for vineyard monitoring over the last two decades) with soil microbial community characterization using next-generation sequencing while also utilizing vineyard canopy images [58] would be vital in predicting an impending poor harvest following severe environmental stress such as drought or heavy precipitation events.

#### 2.4.3. Grapevine Trunk Diseases

Grapevine trunk diseases remain a significant challenge to the wine industry [59,60,61,62]. The most common microorganisms that grapevines tend to be most susceptible to are *Plasmopara viticola* (downy mildew), *Elsinoe ampelina* (anthracnose), *Guignardia bidwellii* (black rot), *Erysiphe neator* (powdery mildew). Interestingly, recent research [63] has demonstrated associations between belowground microbiota *Fusarium* spp. and exacerbating progression of grapevine trunk disease. However, confusion remains concerning the cause and progression of the disease, many of which result in serious infections, loss of yield, or quality. In-depth studies are needed to decipher the interactions between the host grapevine and the diverse fungal communities, some of which may act as opportunistic pathogens under specific conditions. Several studies from the review indicate a growing interest in understanding soil microbial impacts on grapevine health. For example, Liu et al. [35] highlight the importance of microbial diversity in organic versus conventional vineyard management, which can influence disease resistance. Additionally, Mezzasalma et al. [27] and Ramirez et al. [34] discuss pest and disease control practices, emphasizing the need for integrated management strategies that consider microbial interactions. Understanding these interactions could lead to more effective prevention and treatment methods for trunk diseases. Such insights can aid in devising targeted strategies for disease management and prevention. Another particularly intriguing area of research is the role of both asymptomatic and symptomatic grapevines in harboring pathogenic fungi. Understanding the differences in microbial communities between these two states can shed light on the mechanisms underlying the progression of trunk diseases [64].

#### 2.4.4. Bacteriophages for Vineyard Soil Management

Bacteriophages, viruses that infect and replicate within bacterial cells, are abundant in vineyard soils and play a crucial role in regulating bacterial communities. Their ability to selectively target specific bacteria can alter soil microbial composition and dynamics, influencing nutrient cycling, plant health, and likely also grape quality [65]. The field of bacteriophage research in viticulture is still in its infancy [66], and further studies are needed to understand their interactions with bacterial communities and their impact on wine characteristics. For instance, Morrison-Whittle & Goddard [24,31] focused on 26S rDNA to explore yeast diversity, indirectly hinting at the microbial dynamics that bacteriophages might influence. Investigating how phage pressure changes soil bacterial community composition and functions [67] could reveal new strategies for enhancing soil health and grapevine resilience.

## 3. Conclusions

The influence of soil microbial communities on grapevine-associated microbiota is an area of active research and ongoing exploration. Precision viticulture, disease management, bacteriophage research, and next-generation sequencing represent promising areas for future investigation. By delving deeper into these areas, researchers can develop innovative strategies to enhance vineyard health, productivity, and wine quality, contributing to a more sustainable and nuanced understanding of terroir.

## Figures and Tables

**Figure 1 foods-13-02475-f001:**
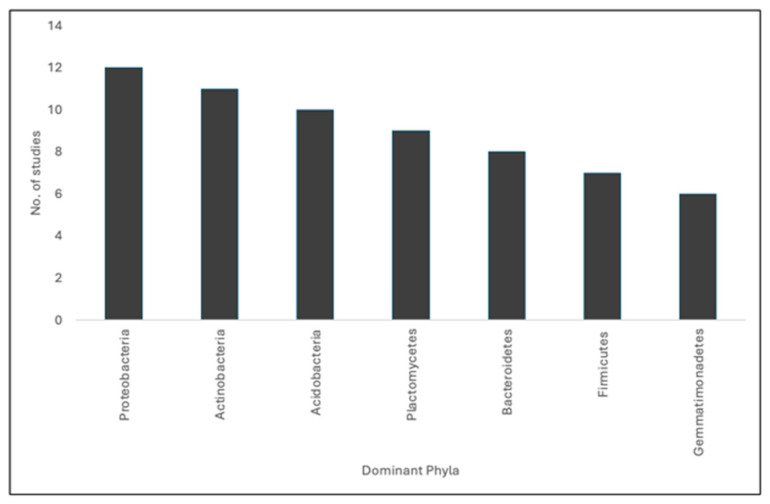
Dominant bacterial phyla in vineyard soil microbial communities.

**Table 1 foods-13-02475-t001:** List of studies on soil microbial communities in vineyards.

Year	Source	Methodology
2015	Burns et al., [23]	16S rRNA
	Morrison-Whittle and Goddard [24]	26S rDNA
	Zarraonaindia et al., [13]	16S rRNA and shotgun metagenomics
2016	Burns et al., [25]	16S rRNA
2017	Casteñada and Barbosa [26]	shotgun metagenomics
	Mezzasalma et al., [27]	16S rRNA and ITS
2018	Hendgen et al., [28]	16S rRNA and ITS
	Chou et al., [29]	16S rRNA
	Wei et al., [30]	16S rRNA and ITS
	Morrison-Whittle and Goddard [31]	26S rDNA
2019	Gupta et al., [32]	16S rRNA and ITS
	Liang et al., [33]	16S rRNA
2020	Ramirez et al., [34]	16S rDNA
	Liu et al., [35]	16S rRNA and ITS
	Oyuela Aguilar et al., [36]	16S rRNA, ITS1, and ITS2
2021	Teixeira et al., [37]	DNA-based assays to detect single nucleotide polymorphisms (SNPs) on three genes of the anthocyanin pathway (UFGT, F3H, and LDOX)
	Rivas et al., [38]	16S rRNA
	Torres et al., [39]	16S rRNA and ITS1
2022	Yan et al., [40]	ITS1
	Geiger et al., [41]	ITS2, ITS4
	Gobbi et al., [42]	16S rRNA and ITS
	Regecová et al., [43]	ITS
2023	Larsen et al., [44]	16S rRNA and ITS1
	Nanetti et al., [45]	16S rRNA

## Data Availability

No new data were created or analyzed in this study. Data sharing is not applicable to this article.

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
