# Peer review of "Soil Microbial Communities and Wine Terroir: Research Gaps and Data Needs"

_foods, 2024, doi:10.3390/foods13162475_

Round 1

Reviewer 1 Report

Comments and Suggestions for Authors

The authors conducted a thorough review of the existing literature on the role of soil microbial communities in vineyards and their impact on various viticultural and oenological factors. Although the manuscript adequately covers the most critical aspects, it lacks a specific indication of the actual results regarding the effects of soil microorganisms on grape health and wine quality. The authors often discuss the aims and rationales without providing the results achieved in the cited studies. For example, sentences in lines 129-132 do not specify the kind of relationship between grape metabolites and soil microbes, and how the soil microbiota influences wine quality. Similarly, lines 166-168 mention the significant influence of soil microbiota on grapevine health and wine sensory properties without detailing the specific microbial genera or species that affect wine quality, or the properties that are indeed affected.

This pattern is consistent throughout the manuscript. It is recommended to include relevant data, also in the form of tables and figures

Author Response

Comment 1: "Although the manuscript adequately covers the most critical aspects, it lacks a specific indication of the actual results regarding the effects of soil microorganisms on grape health and wine quality. The authors often discuss the aims and rationales without providing the results achieved in the cited studies. For example, sentences in lines 129-132 do not specify the kind of relationship between grape metabolites and soil microbes, and how the soil microbiota influences wine quality. Similarly, lines 166-168 mention the significant influence of soil microbiota on grapevine health and wine sensory properties without detailing the specific microbial genera or species that affect wine quality, or the properties that are indeed affected. This pattern is consistent throughout the manuscript. It is recommended to include relevant data, also in the form of tables and figures"

Response 1: The authors thank the reviewer for the comment. In response, the authors have made several insertions of actual taxa and results from the studies throughout the revised manuscript. For coherence and clarity, tracked changes have been highlighted. Specifically, in the tracked changes version of the revised manuscript, the authors have included Figure 1 in line 204 which details the dominant bacterial phyla found in vineyard soil communities in the 24 studies considered between 2015 and 2023. Additionally, the authors have introduced subtitles for coherence as follows: B. Current state of knowledge on the microbiota contribution to wine terroir expression, B.1. The "core microbiome" concept, B.2. Dominant groups in vineyard soil microbial communities and B.3. Contributing factors that link soil-associated microbiota to wine terroir. Notably, the authors introduced section B.1. The "core microbiome" concept, and inserted lines 129-131, 149-152 which highlight actual results of soil microbes found and ideas from studies concerning their linkages to wine quality. Moreover, lines 158-177 were also introduced with actual results from the literature and citing names of microbes of interest within vineyard soils. Within section B.3, subheadings B.3.1. Climate change impacts, B.3.2. Vineyard management practices, B.3.3. Comparative arguments on soil tilling, and B.3.4. Precision viticulture have been introduced. The entire section was re-written for clarity and flow as well (lines 257-350). A new section B.4. Future directions for soil microbial communities and wine terroir research (351-510) which includes 4 sections on next-gen sequencing, precision viticulture, grapevine trunk diseases and bacteriophages follows. We are hopeful that these revisions answer the reviewer's comments. 

Reviewer 2 Report

Comments and Suggestions for Authors

This manuscript delved into the relationship between soil microbial communities and agricultural ecosystems, particularly vineyard cultivation practices, with a focus on their impact on wine quality and regional characteristics. Through methods such as 16S rRNA gene analysis, the geographical differences in bacterial diversity and composition in vineyard soil were revealed, as well as the correlation between microbial communities, metabolites, and fermentation behavior during wine fermentation. Research has shown that the richness and complexity of soil microbial communities have a potential impact on wine quality, and their diversity and composition are influenced by various factors such as grape planting area, variety, growth stage, and land history. Please refer to the following suggestions to further improve and enhance the quality of this manuscript.

1> At line 189, the long-term and short-term impacts of vineyard management on the vineyard ecosystem were mentioned. The following text only introduced the long-term impacts. It is suggested to supplement the short-term impacts.

2> The manuscript simply divided the research into chronological order (2015-2018 and 2019-2023). This classification is not systematic enough, making it difficult for readers to clearly see the inherent connections and differences between different research directions. It can be classified according to research topics such as soil microbial diversity, the relationship between microorganisms and grape metabolites, and the impact of soil depth on microbial distribution. The findings, methods, and conclusions of relevant research can be discussed in detail under each topic.

3> Although this manuscript mentioned research gaps and data needs, the wording is relatively vague and does not specifically indicate which specific areas or problems need to be addressed urgently. In the "gaps and research needs" section, several key and unanswered questions are specifically listed, such as "How does soil microbial community affect grape metabolism pathways and thus affect wine flavor?" and specific data types and analysis methods are required to address these issues.

4> Some paragraphs contain too much content. It is recommended to divide the paragraphs reasonably to ensure that each paragraph revolves around a central idea.

Comments on the Quality of English Language

good

Author Response

Comment 1: At line 189, the long-term and short-term impacts of vineyard management on the vineyard ecosystem were mentioned. The following text only introduced the long-term impacts. It is suggested to supplement the short-term impacts.

Response 1: The authors thank the reviewer for the comment. In response, the authors have revised the section and labeled the subheading "B.3.2. Vineyard management practices" (line 266). Additionally, the section, now revised for coherence and clarity (lines 267-323), highlights both short-term and long-term practices with appropriate references made to the 24 studies considered between 2015-2023. For example, these lines were inserted "The vineyard management techniques discussed in the literature (Table 1) are irrigation management (Aguilar et al 2020; Ramirez et al 2020), canopy management (Ramirez et al 2020), pest and disease control (Regecova et al 2022; Mezzasalma et al 2017), nutrient management (Burns et al 2017; Wei et al 2018), precision viticulture (Teixeira et al 2021), and soil tilling (Hendgen et al 2018)." The authors have also introduced another section "B.3.3 Comparative arguments on soil tilling" (lines 324-337) and "B.3.4. Precision viticulture" (lines 339-350) to expand on this section and highlight relevant monitoring and management of soil microbial communities that might have implications for wine quality. 

Comment 2: The manuscript simply divided the research into chronological order (2015-2018 and 2019-2023). This classification is not systematic enough, making it difficult for readers to clearly see the inherent connections and differences between different research directions. It can be classified according to research topics such as soil microbial diversity, the relationship between microorganisms and grape metabolites, and the impact of soil depth on microbial distribution. The findings, methods, and conclusions of relevant research can be discussed in detail under each topic.

Response 2: The authors thank the reviewer for the comment. In response, the authors have revised the manuscript for coherence and clarity. Specifically, the authors have introduced the following sub-sections under the header B. Current state of knowledge on the microbiota contribution to terroir expression: B.1. The "core microbiome" concept, B.2. Dominant groups in vineyard soil microbial communities and B.3. Contributing factors that link soil-associated microbiota to wine terroir. Section B.3. then expands into the following sub-sections: B.3.1 Climate change impacts, B.3.2. Vineyard management practices, B.3.3. Comparative arguments on soil tilling, and B.3.4. Precision viticulture. The authors have also re-written the final section on the paper which also has new content B.4. Future directions for soil microbial communities and wine terroir research - which subsections: B.4.1 next-generation sequencing, B.4.2.Expanding precision viticulture, B.4.3.Grapevine trunk diseases and B.4.4. Bacteriophages for vineyard soil management. The findings, methods, and conclusions of relevant research have been discussed in detail under each topic. 

Comment 3: Although this manuscript mentioned research gaps and data needs, the wording is relatively vague and does not specifically indicate which specific areas or problems need to be addressed urgently. In the "gaps and research needs" section, several key and unanswered questions are specifically listed, such as "How does soil microbial community affect grape metabolism pathways and thus affect wine flavor?" and specific data types and analysis methods are required to address these issues.
Response 3: The authors thank the reviewer for this comment. In response, the authors have re-structured the entire article as mentioned in response to comment 2 with the headings and sub-headings. Additionally, the authors have highlighted key microbes (lines 129-152) and likely metabolic pathways (such as in lines 169-177) in addition to highlighting speculative aspects of the selected articles from 2015-2023. This validates the objective of this perspective in pointing out research gaps and data needs that warrant future investigation. Another section is 224-235, where the authors point out aspects covered and not covered in greater detail for clarity. This is repeated throughout the manuscript in the new and revised sections. Please refer to the tracked changes version of the attached manuscript. 

Comment 4: Some paragraphs contain too much content. It is recommended to divide the paragraphs reasonably to ensure that each paragraph revolves around a central idea.
Response 4: The authors thank the reviewer for this comment. In response, the authors have restructured the entire manuscript with introduced headings and sub-headings to breakup the paragraphs and also remove redundant information. Please refer to the tracked changes version of the attached manuscript for the revisions made. 

Round 2

Reviewer 1 Report

Comments and Suggestions for Authors

The paper has been significantly improved after addressing the main issues. In my opinion, it is nearly ready for publication, with only a couple of minor issues remaining, such as the missing legend for the added figure and the reference in the text.

Author Response

Comment 1: Missing legend in the figure
Response 1: Thank you for the comment. We have added the legend for the figure.

Comment 2: Reference in text
Response 2: Thank you for the comment. We have reviewed the entire manuscript for in-text citations and associated bibliographies. Kindly find attached the revised manuscript with tracked changes. 
